# Efficiency of In-Store Interventions to Impact Customers to Purchase Healthier Food and Beverage Products in Real-Life Grocery Stores: A Systematic Review and Meta-Analysis

**DOI:** 10.3390/foods10050922

**Published:** 2021-04-22

**Authors:** Helena Slapø, Alexander Schjøll, Børge Strømgren, Ingunn Sandaker, Samira Lekhal

**Affiliations:** 1Department of Behavioral Sciences, Oslo Metropolitan University, P.O. BOX 4 St., Olavs plass, N-0130 Oslo, Norway; bstromgr@oslomet.no (B.S.); isandake@oslomet.no (I.S.); 2GreeNudge, Tordenskioldsgate 2, N-0160 Oslo, Norway; samlek@siv.no; 3Consumption Research Norway, Oslo Metropolitan University, P.O. BOX 4 St., Olavs plass, N-0130 Oslo, Norway; alexan@oslomet.no; 4Morbid Obesity Centre, Vestfold Hospital Trust, Halfdan Wilhelmsens allé 17, 3116 Tønsberg, Norway

**Keywords:** consumer behavior, healthy and nutritional food choices, food choice motivations, choice architecture, nudging, food environment, interventions, obesity, systematic review

## Abstract

Grocery stores are important settings to promote healthier food and beverage choices. The present paper aims at reviewing the effectiveness of different types of in-store interventions and how they impact sales of different product category in real grocery stores. Systematic search was conducted in six databases. In-store interventions were categorized according to the framework by Kraak et al. (2017) into one or more of eight interventions (e.g., place, profile, portion, pricing, promotion, healthy default picks, prompting and proximity). This systematic theme-based review follows the preferred reporting items for systematic reviews and meta-analyses (PRISMA) data screening and selection. Thirty-six studies were included in the qualitative synthesis and 30 studies were included in the meta-analysis, representing 72 combinations of in-store interventions. The analysis demonstrates that interventions overall had small significant effect size (ES) using Cohen’s *d* on food purchase behavior (*d* = 0.17, 95% CI [0.04, 0.09]), with largest ES for pricing (*d* = 0.21) and targeting fruits and vegetables (*d* = 0.28). Analysis of ES of in-store interventions show that pricing, and pricing combined with promotion and prompting, effectively impacted purchase behavior. Interventions significantly impacted both sales of healthy and unhealthy products and significantly increased sales of fruits and vegetables, healthy beverage and total volume of healthy products. Results should however be interpreted with some caution, given the relatively low quality of overall evidence and low number of studies and observations for some types of intervention. Further research exploring impact on different in-store interventions and targeting especially unhealthy products are needed.

## 1. Introduction

Unhealthy diets are among the most important risk factor for illness and reduced quality of life worldwide [1]. An increase in consumption of foods high in sugar and saturated fats, and low consumption of fruits, vegetables, whole grains, nuts, seeds and seafood, coupled with a lack of physical activity, are key factors explaining the rise in chronical diseases, overweight and obesity worldwide [2,3]. Purchase and consumption of unhealthy diets, especially low fruit and vegetable intake is strongly patterned by socioeconomic status (SES) [4,5,6,7,8].

Numerous interventions aiming to change peoples’ food habits through changing beliefs and attitudes via information and education campaigns have been tested. These interventions build on economic theories and models of rational decision-making, assuming that human choices are reason based, rational and logical [9,10]. People are assumed to seek information on the quality and cost of the feasible options and systematically use this information to maximize their utility—that is, make the choice that is in their best interest [11]. As of today, these interventions alone do not seem to have solved problems of food consumption, because unhealthy diets and obesity seems to be an increasing problem [2,12].

Recently, more attention from a policy perspective has been given to how the retail environment can be designed to encourage healthier food purchases [13,14,15]. The majority of foods and beverages in high-income countries are bought in grocery stores and supermarkets [16,17]. Therefore, effectiveness of in-store interventions are of particular interest [18]. In-store interventions may be particular suited to impact people from lower socioeconomic groups, as research has shown that this group may be less impacted by information and education campaigns outside stores [19,20,21].

Grocery purchases are often not planned in detail and are expected to be strongly habitual. The principle within behavioral economics suggests that habits can be improved by changing the environment within which people make choices; otherwise known as choice architecture or nudging [22]. According to Sunstein and Thaler [23], choice architecture impacts a decision by simplifying the presentation of options, by automatically evoking particular associations, or by making one option more easier to choose than the alternatives. Hollands et al. [9] define choice architecture as “interventions that involve altering the properties or placement of objects or stimuli within micro-environments with the intention of changing health-related behavior” (p. 3). Nudging represents only one form of choice architecture, whereas use of price incentives or limiting access to unhealthy options are also considered choice architecture tools [23].

This review contributed with new knowledge to the existing literature in at least three ways. First, this review aims to include a wide range of in-store interventions. Previous reviews have often only looked at either typical marketing interventions or choice architecture interventions in grocery stores [15,24,25,26,27,28]. For instance, Cadario and Chandon [15] only included pure nudge interventions, and did not include price interventions. We believe it to be particularly important to include price interventions made within stores as price campaigns, as they often are used in combination with other interventions, as promotion and prompts, and are likely to have an influence on consumers behavior. Second, previous reviews that did used a wider range of interventions were not limited to experimental studies and effect size of interventions was not evaluated [28,29,30]. This review will only include experimental studies and studies conducted in real grocery storers and supermarkets. In addition, we will calculate the effect size of interventions. Third, other reviews have included studies from stimulated or laboratory settings (Bauer and Reisch [12]; Cadario and Chandon [15]; Escaron, Meinen [25]; Glanz, Bader [26]; Hartmann-Boyce, Bianchi [30] and Liberato, Bailie [31]. As previous studies have shown, people may act differently when they know that they are being monitored in laboratory settings, we see the need to evaluate interventions effect for only real-life settings [15,30,32].This is especially important for the evaluation of pricing interventions [33]. Few previous reviews have focused exclusively on the grocery store setting, instead included studies performed also in work cafeterias, school cafeterias and corner stores [15,26,31]. These settings serve a different purpose than grocery stores, as food is consumed right after purchase and these setting account for a smaller portion of food and beverage purchases than grocery stores.

The objective of this systematically theme-based review and meta-analysis is to compare the effectiveness of in-store interventions to encourage customers to purchase healthier foods and non-alcoholic beverages in real physical or online grocery stores. Additionally, the study will evaluate the variability in effectiveness of different combinations of in-store strategies and different targeted food category. A systematic literature review was appropriated based on the widely used theories, methods and constructs, to identify the research gap in current research and to offer suggestions for further research, as seen in more recent review articles [34,35,36,37].

The remainder of this article is structured as follows. Section 2 illustrates the methodology, describing eligibility criteria, search strategy, data collection, data extraction, method for risk of bias, statistical analysis and quality of evidence. Section 3 presents the results for the literature review, including a thematic synthesis of included studies, risk of bias and results of the meta-analysis. Section 4 discusses the findings, strengths and limitations of the paper’s analysis. Section 5 gives the conclusions and suggestions for further areas of research.

## 2. Method

This review follows the preferred reporting items for systematic reviews and meta-analyses (PRISMA) statement [38]. The protocol was submitted to PROSPERO on 13 November 2020 (see the Appendix A for the complete protocol).

### 2.1. Eligibility Criteria

This review included randomized controlled trials (RCT), controlled before and after studies (CBA) or interrupted time series (ITS). The recommendations by the Cochrane Consumers and Communication Review Group on which non-randomized studies to include were followed when addressing questions of effectiveness of interventions [39]. We included CBAs with at least two control stores and two intervention stores, similar key characteristics of control and intervention groups, and comparable timing of study-periods. ITS studies were included if they had a clear description of when the intervention took place, and at least three datapoints before and after the intervention [39]. There were no restrictions on population types as long as they were targeted within grocery stores. The setting was real physical grocery stores or real online grocery stores.

The choice architecture interventions included in this review comprise any in-store intervention that involved altering the properties or placement of objects or stimuli within microenvironments with the intention of changing health-related behavior, as defined by Hollands et al. [40]. Additionally, interventions had to take place in stores, be designed to improve customer’s food purchases and target customers at individual-, store- or products-level. Interventions were for the narrative synthesis and meta-analysis categorized according to the marketing mix and choice architecture framework by Kraak et al. [41]. Price change were included in the review if the interventions were seen by customers as something that the store owners had initiated. Pricing studies based upon external initiatives, such as external taxes, subsidies or price change provided by others than store owners, like part of the community health programs or by insurance companies, are not included. In order to be included in the review, the study had to include an outcome measure of change in objective purchase behavior, either through sales data or by asking customers about their purchase right after purchase. Targeted product had to be food or a non-alcoholic beverage.

### 2.2. Information Source and Search Strategy

Search was done in collaboration with a science librarian with expertise in systematic review searching, at OsloMet. The search was conducted in six databases: Cochran’s, PsychInfo, EconLit, Medeline, Scopus and Web of Science on 24 April 2020. Specific search terms were developed according to the SPICE (setting, population, intervention, comparison and evaluation framework) [42]. The search was restricted to the English language and studies published in peer-reviewed journals. Appendix A shows how search words were developed and the search words included in all databases are shown in Appendix A. Search strategy included a range of broad search terms, related to different types of in-store interventions, as the research area is broad and interdisciplinary. Additional articles were identified by analyzing the identified reviews from the literature search and from reviews identified through snowballing.

### 2.3. Data Collection

Two authors (HS and SL) independently screened titles, abstracts, and full text, for all studies meeting the inclusion criteria, were thereafter obtained. Any disagreements were resolved by discussion. Only where both authors agreed, the study was included for full text screening and later included in this systematic review.

### 2.4. Data Extraction

For the narrative synthesis, the following information was extracted from each study: year of publication, country, setting, study design, type of intervention, duration, sample size, targeted product, population type, outcome measures and main findings. Type of in-store intervention was categorized according to the framework by Kraak et al. [41], i.e., placed into one of more of eight strategies (e.g., place, profile, portion, pricing, promotion, healthy default picks, prompting and proximity). Since the framework by Kraak et al. [41] was created to impact customers in restaurants, we made some adaptions for the framework to fit for a grocery store setting. Table 1 displays the adapted version.

Classification of targeted food or non-alcoholic beverages were classified as healthy or unhealthy based on the World Health Organization (WHO) recommendations for healthy diets [43]. Therefore, an increase in the purchase of fruits and vegetables, whole grains, high fiber products, water or other non-sugar beverage products were categories as targeting increases in healthy food purchases. Reduction in sales of high fat/saturated-fat/sugar/salt/calorie products and sugar-sweetened beverage products were categories as targeting unhealthy food purchase.

### 2.5. Risk of Bias within Individual Studies

To assess and report on methodological risk of bias for RCT and CBA studies, we used the Cochrane risk-of-bias tool [44] and the guidelines of the Cochrane Consumers and Communication Review Group [45]. The individual studies were rated as having either low, unclear or high risk of selection bias, performance bias, detection bias, attrition bias, reporting bias and other biases. CBA studies were rated against the same criteria as RCT studies but were regarded as having high risk of selection bias due to not being randomized [44]. Assessment of the quality of the included ITS studies was based on a Cochrane EPOC Review Group tool [39], which recommends the following individual elements for ITS: intervention independence of other changes; prespecification of the shape of the intervention effect; likelihood of intervention affecting data collection; blinding (participants and personnel); blinding (outcome assessment); completeness of outcome data, selective outcome reporting and other sources of bias (e.g., if seasonality not accounted for).

Two authors (H.S. and H.B.) independently assessed the risk of bias of all included studies, with any disagreements resolved by discussion to reach consensus. We contacted study authors for additional information about the included studies, or for clarification of the study methods, if required.

### 2.6. Statistical Analysis

In order to evaluate and compare heterogeneous outcomes, results calculated effect size (ES) using Cohen´s *d*, with a general interpretation for the social sciences of *d* ≥ 2 being a small ES, *d* ≥ 5 being a medium ES and *d* ≥ 8 being a large ES [46]. A random effects model was used for the analysis, which assumes that the effects being estimated in the different studies are not identical, but follow some distribution [44].

Homogeneity tests were conducted using Cochrane Q for between-study heterogeneity, and I^2^ statistics for evaluating magnitude of heterogeneity. I^2^ values of 25%, 50% and 75% were regarded as low, moderate and high heterogeneity, respectively [47]. Publication bias was assessed by looking for asymmetry in the funnel plot and the Egger test.

Subgroup analysis was performed for the type of intervention and type of targeted product separately, as many studies mentioned separate results for different combinations of interventions and targeted products. When multiple dependent variables were measured in studies (as self-reported consumption and sales data), only one dependent variable was included in the meta-analysis—usually sales, which is the most objective one. Additionally, if outcome was reported in dollars spent and grams purchased, only grams purchased were used for the ES calculation. If studies reported effect separately for different categories (e.g., fruits and vegetables separately) and total for a category (e.g., total fruits and vegetables), we calculated ES for the total category. When effect was reported for multiple time periods, we calculated ES by comparing the control condition compared to the time period right after the end of the intervention (i.e., at post intervention and not at follow-up intervention).

Descriptive analyses on the narrative synthesis were performed using IBM SPSS v. 21. Meta-analysis was conducted with the help of the Meta-Essential Software [48]. Effect size and standard error was calculated with the help of the two web-based calculators developed for meta-analysis [49,50]. All meta-analysis calculations were plotted into Excel using the Meta-Essentials Workbooks for Meta-Analysis [48,51].

### 2.7. Quality of Evidence

The grading of recommendations assessment, development and evaluation (GRADE) was used to assess the overall quality of evidence for the included RCT studies. Evidence was downgraded from “high quality” by one level in case of serious (or by two for very serious) study limitations (risk of bias), indirectness of evidence, serious inconsistency, imprecision of effect estimates or potential publication bias.

## 3. Results

### 3.1. Study Selection

Figure 1 shows the information flow of the scanning process. A total of 3108 unique citations were identified through the initial search, whereof 2653 were excluded because of irrelevant content or duplications. The full text of 874 articles was assessed, with 419 of these not meeting the inclusion criteria. Finally, 36 studies were included in the systematic review and 30 studies contained sufficient information to be included in the meta-analysis. See Appendix A for more information on search strategies and Appendix A for complete reason list of excluded articles and reason for exclusion.

The most common reason for the exclusion of studies from the meta-analysis were the lack of data necessary to calculate effect size, i.e., sample size for the intervention and control conditions or means and standard deviations. Many of the included studies evaluated effect for intervention on different product categories separately. Most of the included studies used more than one combination of grocery store interventions (e.g., evaluating effect of promotion and price interventions or promotion alone) and dependent variable (e.g., effect measured on fruits and vegetables and fat purchase). For the estimate of the effect size, each combination of intervention or target product category was registered separately. Doing this for all 30 studies in the meta-analysis (*n*), we ended up with 72 interventions studies (k).

### 3.2. Study Characteristics

Key characteristics of included studies are summarized in Table 2. Of the 36 included studies in this review, 22 were RCT, six were CBA and eight were ITS. The studies were published between 1982 and 2020. The majority of studies took place in North America (21 studies), ten studies in Europe, three studies in Oceania and two studies in Asia (Singapore). Eight studies targeted low-income customers, either by doing an intervention in low-income neighborhoods [52,53,54,55,56,57] or by targeting overweight or obese customers within stores [58,59]. Of the studies 33 had physical grocery stores as a study setting, with only three conducted in an online grocery store setting. The number of included stores (control and interventions stores) ranked from 1 to 2000 stores, with an average number of control stores of 8 and an average number of intervention stores of 75. Study duration (baseline and intervention phase) ranged from one week to three years with an average of 24-weeks. In 28 studies, changes in food choices were assessed through sales data, as units sold collected through check-out transaction data or loyalty card databases. Number of observations ranged from 47 to 1,920,000 with an average of 125,366 observations. The frequent use of transaction data from stores as outcome measure explains the extremely high number of observations in some of the studies.

In all studies products were categorized as healthy or unhealthy in accordance to the WHO recommendations for healthy diets [43]. In almost half of the studies (16 studies), the goal was to increase sales of healthy products and the most targeted product category was fruits and vegetables, which was covered in 15 studies. Studies defined healthy products as fruit or vegetable [52,53,54,59,60,61,62,63,64,65,66,67,68,69,70], products with high nutritional “grade” (e.g., “3-star”or “green” rating) [57,71,72,73,74,75,76,77,78], high fiber [61,62,70], low fat products [56,79,80,81], low-calorie snacks [81] or healthy beverages (e.g., water or diet soda) [56,63].

Eight studies targeted reduction in unhealthy food products, with reduction in fat (total fat or saturated fat) and sugar soda being the most targeted outcomes [55,56,61,62,63,66,70,75,76,82,83]. Studies. Defined unhealthy products as high fat or saturated-fat products [56,61,62,66,70,82,83,84], products with low-nutritional rating (e.g., unstarred or “red” rating) [73,74,77,78], sugar soda [56,63] and unhealthy snacks (e.g., chocolate, chips or cake) [58,85,86]. Reduction in total calories purchased [85,86] and reducing sales of the most unhealthy products within certain categories [77] was also defined as targeting unhealthy products.

In 12 studies the goal was to increase both sales of some healthy products and reduce sales of other unhealthy products. The most common combination was an increase in sales of high fiber products, fruit and vegetable and reducing in sales of high fat products [61,62,70].

In-store promotion was the most frequently studied intervention (22 studies). In six studies promotion was tested together with pricing and together with proximity in six other studies. The most common promotion strategies were giving out product samples, food demonstrations or giving customers education sessions about the health benefits of targeted products. Use of prompts were studied in 11 studies, including the two where it was studied together with promotion. Information rich prompts, as nutrition labels and low-calorie labels were the most tested prompt. The guiding star labeling system (a program that indicated products with a multiple-level summary prompt rating product from 0 to 3 stars and that was developed for and introduced in one American supermarket chain) was evaluated in three of the studies [72,73,74]. Pricing was studied in nine studies, mostly in combination with promotion. In all of these studies, the price for healthy products was reduced. Use of proximity was studied in seven studies, mostly in combination with promotion (six studies). Healthy default picks and proximity were studied in one study each. None of the included studies used portioning or place strategies.

### 3.3. Risk of Bias

Risk of bias for the 22 RCT are shown in Figure 2, for the six CBA studies and for the eight ITS in Figure 3 and Figure 4. The full description of the authors’ judgment of risk of bias for each domain and support for judgment is given in Appendix A.

For the RCT studies, most of the studies were rated as having unclear risk for selection bias (random sequence generation and allocation concealment) because of an inadequate description of the randomization method. One study was rated high risk of allocation concealment, since researchers told participants about the study objective and since outcome at the same time was measured by participants answering a questionnaire about food purchases [79]. Blinding (participants and personnel) was ensured in almost all studies, except three [52,61,64], where participants were told about the study objective and outcome at the same time was measured through self-reported measurements. Blinding of the outcome assessment was ensured in most of the studies as outcome was measured objectively through sales data of whole stores, or by getting access to the shopper’s loyalty card database linked to individual sales. Most studies were regarded as having low risk of attrition bias. However, four studies were rated as high risk of attrition bias since they had higher dropout in their intervention group than in their control group [59,62,63,66]. Most of the studies were rated as having unclear risk of reporting bias. Only one study was rated as high risk of other biases [63]. The reason was that the intent-to-treat analysis was not reported as prespecified in the protocol. We believe this to be of importance since if reported, this may have impacted effect size calculations and given us a more reliable estimate of true treatment effectiveness [63].

All six CBA studies were rated as high risk of selection bias due to a lack of randomization. For most of the studies, determination of control and intervention stores was based on geographic location of the stores and not randomization. All studies were blinded (participants and personnel) and four were triple blinded, as outcome measurement was objective transaction sales data. Two studies were judged as high risk of detection bias because outcome was measured through telephone interviews [75,81]. All studies were rated as low risk for attribution bias. Insufficient information was available to assess whether any important risk related to selective reporting exist in most of the studies. For one study selection bias was judged as high since outcome was only measured for some products, while interventions was on all healthy products [75]. Four of the studies were rated as unclear for other biases, based on information that baseline and control stores had noticeable differences in characteristics of customer groups in terms of racial and ethnic composition, but it was not clear this may impact evaluating or the effect of intervention.

For the eight ITS studies three other risk of bias assessments were made. These three assessments were: intervention independent of other changes, whether the shape of intervention effect was prespecified and whether intervention would affect data collection. In one study renovations took place during the intervention period but the authors did not state what exactly these renovations included, hence the study was rated high risk for intervention not being independent of other changes [68]. For all studies the expected shape of intervention was prespecified meaning that interventions were expected to improve sales of healthy products. For all studies the data were collected from routine sources and we considered the studies to be at low risk of that intervention affect data collection. All studies ensured triple blinding (participants, personnel and outcome assessment). None of the studies had problems with incomplete outcome data. Low risk of selective outcome reporting was ensured in six studies, and it was unclear for two studies. Five studies were rated as having low risk for other biases since they accounted for seasonal trends or other large differences in control and intervention conditions. This was especially important since baseline phase and intervention phase often were during different months of the year, likely to impact food purchase behavior. In Patterson et al. [77] the authors state that large differences in socioeconomic status between customers at the control and intervention sites exist, leading to a lack of comparability between the two stores. Therefore Patterson et al. [77] was rated as high risk of bias under “other biases”.

It is worth noting that Cochrane risk-of-bias tools [39,44] does not specifically evaluate sample size, study duration or the quality of the intervention. Each of these are important considerations, which is why our Results section focused heavily on these aspects of study quality. This information is likely to be critical in order to evaluate the results of the studies but can also incrementally improve study quality. Interventions using signs are in particular likely to be influenced by the quality of the graphics used.

### 3.4. Results from Meta-Analysis

Effect size calculations was based on 72 different effect size predictions (k) from the 30 studies that were included in the meta-analysis. The analysis demonstrates that in-store interventions overall had a small, significant effect on purchase behavior *(d* = 0.17, 95% CI [0.04, 0.09] and PI [−1.02–1.35]). This is shown in Figure 5.

The overall between-study heterogeneity was considerable (Q-test *p* < 0.01, *I*^2^ = 100%). Virtual inspection of the funnel plot provided mixed evidence for publication bias. However, Egger´s test did not indicate statistical evidence for publication bias (*p* = 0.89) (see the Appendix A for funnel plot and Eggers’s regression).

#### 3.4.1. Analyses by Type of In-Store Intervention

Table 3 shows the frequency of interventions, ES and confidence intervals for the different type of intervention and combinations of interventions.

Promotion combined with pricing was the most common intervention (*k* = 17) and had a significant effect on food purchase (*d* = 0.21, CI [0.08, 0.33]). These studies most commonly evaluated the effect customers education sessions together with coupons or price reductions for healthy food products. Interventions looking at promotion (*k* = 15) also had a significant effect on food purchase (*d* = 0.10, CI [0.02, 0.18]), but the effect was lower than when combining promotion with a pricing intervention.

Prompting were all different nutritional labels on products or shelf and had an overall significant effect on purchase behavior (*k* = 12, *d* = 0.14 and CI [0.09, 0.19]). Four interventions looked at the effect of the guiding star labeling system [73,74] and one found a significant effect on food purchase behavior (Rahkovsky et al. [73], 2013 *d* = 0.19 and CI [0.11, 0.27]).

Price reductions on healthy products had the largest effect on food purchase (*k* = 11, *d* = 0.42 and CI [0.02, 0.82]). Price was reduced by 12.5%–50% and/or by giving customers a $25 gift card for not purchasing unhealthy products. The two interventions with the largest ES both used 50% discounts on fruit and vegetables (Geliebter et al. [59], *d* = 1.35, CI [1.34, 1.37]; Polacsek et al. [54], *d* = 1.62 and CI [1.56, 1.68]).

Healthy default picks also had a large and significant ES (Huang et al. [83], *d* = 0.52 and CI [0.30, 0.75]). However, this was only based on one intervention study [83] and we cannot draw strong conclusions about the effect of healthy default picks. Only one intervention study also evaluated the effect for the combinations: pricing and prompting [55]; promotion and prompting [82] and proximity [68]. Number of interventions was therefore too limited to draw strong conclusions.

#### 3.4.2. Analysis by the Targeted Product Category

Table 4 shows the frequency of interventions, ES and confidence intervals for the subgroup analysis for the targeted product category.

Most intervention studies targeted sales increase of healthy food and beverage products (*k* = 46). Overall combined ES for healthy products was significant (*d* = 0.19 and CI [0.09, 0.29]. Most studied category was fruits and vegetables with a significant ES (*k* = 21, *d* = 0.28 and CI [0.08, 0.48]) [52,53,54,59,61,62,63,64,65,66,68,69]. Additionally, interventions targeting increasing purchase of a healthy beverage as water and diet soda (*k* = 10) [56,63] had a significant ES but the effect was small (*d* = 0.01 and CI [0.01, 0.02]). Seven of the eight interventions that measured effect for increasing sales of total volume of healthier products used prompting interventions on many products at the same time. Overall ES for these studies was significant but small (*d* = 0.16 and CI [0.14, 0.17]) [57,71,73,74,84]. This analysis includes too few interventions targeting high fiber products, low-fat cheese and low calories snacks for us to make any strong conclusions about the effect of in-store interventions on those targeted products.

In 26 intervention studies the goal was to reduce customers’ purchase of unhealthy products and combined ES considered small and significant (*d* = 0.11 and CI [0.03, 0.19]) [55,56,58,61,62,63,66,70,73,74,82,83,84,85,86]. Fat purchase (*k* = 7) and reducing purchase of an unhealthy beverage such as sugar soda (*k* = 6) were the most common targeted unhealthy products [55,56,61,62,63,66,70,82,83]. None of the studies that evaluated the effect of interventions on different unhealthy categories showed a significant impact on purchase.

## 4. Discussion

### 4.1. Summary of the Main Findings

This analysis demonstrated that in-store interventions, which change the environment within which people make choices, successfully improved purchase habits of customers in real grocery stores. In-store interventions seem to be especially impactful for increasing sales of healthy products, and these studies significantly increased sales for fruit and vegetables, water and diet soda. The effect on sales of unhealthy products is less clear and needs further investigations.

In-store interventions should be considered as additional strategies to more traditional and general policy strategy, as education and restrictions, to promote healthy food consumption. It should however not be seen as strategies that replace stricter public policies, but as a strategy that is available within the grocery store.

The overall effect size of 0.17 was considered small *d* ≥ 2 [46], but within a similar range as previous reviews have found for choice architecture interventions of food choices in other settings [15,90,91]. Even small changes in dietary habits have the potential to improve general health tremendously [1,92,93]. To get a more tangible description of that this means in a public health perspective we computed what this would translate to for daily calorie intake of adults. Inspired by Cadario and Chandon [15] we calculated what an effect size of 0.17 would mean for the energy intake of an average American adult. Given that that average calorie intake of Americans adults amount to 2880 kcal per day [94], this translates to a calorie reduction of 476 kcal per day. This is the same calorie content as a chicken burger at McDonalds.

Based on the analyses it is reasonable to conclude that pricing (*d* = 0.40) and the combination of promotion and pricing (*d* = 0.21) seem to be the most effective strategy. Somewhat surprisingly the effect of pricing was larger in studies where it was tested alone, than then used together with the promotion. The two intervention studies that showed large ES (*d* > 0.8) used 50% discounts on fruit and vegetables (Geliebter et al. [59], *d* = 1.35; Polacsek et al. [54] *d* = 1.62). A review by Thow et al. [95] evaluating the effect of price reduction on healthy foods found results consistent with our review. Both conclude that pricing is likely to be effective in altering customers purchase behavior.

Analysis for targeted products show that fruits and vegetables was the most targeted food category with a significant ES (*d* = 0.30). This was the same ES of nudge intervention for fruits and vegetables found in the systematic review by Arno and Thomas [91]. Analysis also shows a small significant ES when studies targeted an increase in healthy beverage sales (water and diet soda) (*d* = 0.01), increased total volume healthy products sold (product labeled as healthy) (*d* = 0.16), increase in low-fat cheese (*d* = 0.19) and reduction in sales of fat (total fat and saturated fat) (*d* = 0.10). For the other targeted product, the analysis shows a non-significant effect.

It is important to note that the overall number of observations is limited, and few studies evaluated the effect of some types of interventions (e.g., only one study each looked at healthy default picks; proximity; combining promotion and prompting and combining profile and proximity). Additionally, no studies evaluated the impact of portioning or place strategies.

### 4.2. Quality of Evidence

Using the GRADE approach to assess the overall quality of the evidence, we rated the quality from the RCT studies as very low (see Table 5). This was due to the fact that (1) most of the RCT ensured blinding, but that most studied lacked information about randomization process, which made for two downgradings on “Risk of Bias”, (2) there was serious “Inconsistency” due to heterogeneity (*I^2^* = 99.8) and (3) “Imprecision” due to wide CIs (ES = 0.18 and CI [0.10, 0.26]). Finally (4) “publication bias” was not suspected (Orwins Failsafe-N = 166; Trim and Fill [44] suggest 13 imputed studies changing the ES from 0.18 to 0.24). For the six CBAs were overall considered the quality of evidence to be of low quality. This was mainly due to (1) the high risk from a lack of randomization and insufficient information to assess whether an important risk of selective reporting was present. Furthermore, there was a serious risk of “Imprecision” due to wide prediction intervals (*d* = 0.14 and CI [0.01, 0.30]). The eight ITS were well conducted and met most of the criteria for a low risk of bias in the assessment. However, seasonality was addressed inconsistently between studies and different types of analyses were used in order to address this. Therefore, overall quality of ITS was judged as low quality. Based on the GRADE approach, we concluded that overall quality to be a moderate true effect of in-store intervention may be moderately different from the estimate of effect.

### 4.3. Strengths and Limitations

This is a systematic review with a meta-analysis. This approach may increase the objectivity and transparency of the selection and analysis of the articles. It also strengthens the results presented. It also implies that studies that did not use the exact search terms considered are not included. We are aware that we may have excluded strategies that were not conceptualized as marketing, choice architecture or nudge interventions. We minimized possible selection biases in the review process by using a comprehensive search strategy to identify studies and, wherever possible, independently selecting and appraising the studies. In addition to searching journal in multiple electronic databases, we also searched for previously published review articles on similar subjects. Restricting the search to RCT, CBA and ITS, minimizes confounding, making us more confident in our results than some previous reviews. For all the included studies, sales data or self-reported behavior right after the fact was the most used outcome measurement. This avoids the problem that self-reported attitudes, intentions or past behavior is open to bias, such as socially desirable responding. Sales data as outcome measurement from real stores is therefore considered powerful for evaluating purchase behavior [84,96].

The scope of this review made us able to take a broad approach to the research available on the effect of in-store interventions, including a broad set of interventions, settings, population groups and aims. This however, lead to a large diversity in intervention type as they were named differently in different studies, which could have caused problems with heterogeneity. In the planning of this review, we therefore made a pragmatic decision to categorize intervention according to a modified version of Kraak et al. [41] framework, which was relatively straightforward for some interventions, but less straightforward for others. When we were unsure about categorization, we were guided by the existing literature [25,26,29,30,31]. We are aware that other researchers may had categorized interventions differently, particularly prompting and promoting. In this review we did however not run into any strong disagreements about categorization, and any disagreements were resulted through discussion.

An important limitation of this study was the relatively low number of included studies. A large number of studies identified in the systematic search were not included in the review. One of the most common reasons for exclusion was that the study design was non-experimental and/or that the study was not conducted in real stores. The reason why few studies met inclusion criteria may be that experiments that include a suitable control contrition and ensure randomization may be harder to conduct in real-life settings, especially when the study is conducted in collaboration with store owners.

In addition, as many as seven studies of the included studies did not contain enough statistical information to be included in the meta-analysis. This could lead to different conclusions about the effect of different interventions. On the other hand, we found no reason to assume that the effectiveness of in-store interventions in studies not included in the meta-analysis would be different from those included.

Another potential problem is the publication bias, as articles with significant results are more likely to be published than those with non-significant results [97]. Therefore, the effect of in-store interventions may be smaller in real life than indicated by this analysis. As previously noticed, the Eggers test did however not indicate statistical evidence for publication bias.

## 5. Conclusions

Results in this systematic review and meta-analysis demonstrated that inside grocery store interventions—particularly when targeting healthy products—significantly encouraged healthy behavior. The effect size of the intervention overall was considered relatively small. However, given the prevalence of unhealthy diets as a health issue, and the important role grocery stores play in shaping food and beverage purchase behavior, we believe that in-store interventions still play an important role. We believe that the design of grocery stores could and should be considered as a part of a more general policy strategy that seek to promote healthy food consumption and prevent obesity.

This work provides some evidence that price discounts alone or in combination with in-store promotion seem to be the most promising in terms of increasing sales of healthy food options. The effect size of in-store interventions was larger when targeting healthy products (e.g., fruits and vegetables), than when targeting unhealthy products.

The findings in this paper offers insights into further research needs to strengthen the ability to make general recommendation from the data. In order to fully demonstrate the effectiveness of in-store interventions it would be best to perform a meta-analysis with more studies of higher quality. In order to make this possible, new studies should report on sample size, means and standard deviations to make effect size calculations more precisely. Studies should be designed to ensure randomization, include a control condition and seasonality should be addressed more consistently. In addition, new measurement techniques that make it possible to track the food purchase of individual people, instead of using self-reported technique would enhance the quality of evidence.

Further research is needed on how in-store interventions can be used to reduce sales of unhealthy products. For instance, no studies have looked at the effect of increasing pricing of unhealthy products or giving them less attractive placement in stores. Further studies should target especially high saturated-fat and high sugar products. Interventions that lead to a reduction in consumption of unhealthy products can be good for business if it leads to attracting new consumers and if people like being nudged towards healthier choices. Especially if healthy have a better markup than (cheaper) unhealthy products. For some interventions we also had to few observations to say anything certain about their effects, including healthy default picks, portioning, proximity and place and profile. Therefore, more research is needed on how these interventions impact purchase behavior alone and in combination with each other or outside store interventions.

Further research should look into what aspects of in-store interventions seem to be the most effective across consumer groups, regions, countries and grocery store settings. Most of the included studies in this review were conducted in North America, giving a good theoretical framework for future work in other high-income countries. However, to ensure generalizability of findings, researchers should conduct studies in more diverse countries and regions and include subgroup analysis of the in-store interventions effect on different population groups. Subgroup analysis in the included study by Huang et al. [83] provided evidence that in-store interventions had a greater effect on people with high BMI. Since populations with higher risk for unhealthy diets and obesity should be targeted, we welcome further research that tests the effect of in-store intervention on lower income and high obese populations. Further research in online grocery stores are needed since they are increasing their market shares worldwide and becoming an increasingly important part of the grocery’s future.

In-store interventions described in this review, followed the definition of choice architecture interventions defined by Hollands et al. [9]. This meant that interventions intent to make it easier for customers to make healthier choices, without limiting the access to unhealthy products. Further research should evaluate how people feel and if they liked being nudged towards healthier options. If further studies evaluate impact on customers satisfactions and loyalty, it is likely that grocery store owners will adapt them because it may give indications that nudging customers towards healthier choices are compatible with commercial goals.

## Figures and Tables

**Figure 1 foods-10-00922-f001:**
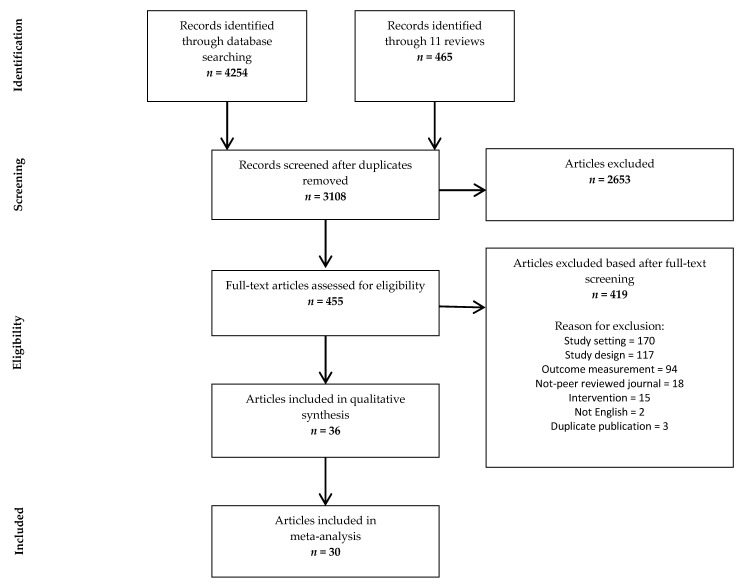
PRISMA diagram of study flow [38].

**Figure 2 foods-10-00922-f002:**
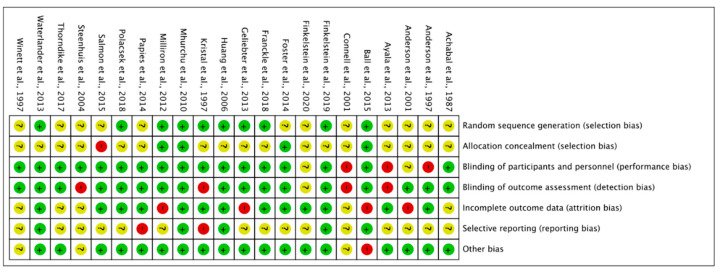
Risk of bias summary: review of authors’ judgments about each. Risk of bias item for RCT studies (*n* = 22).

**Figure 3 foods-10-00922-f003:**
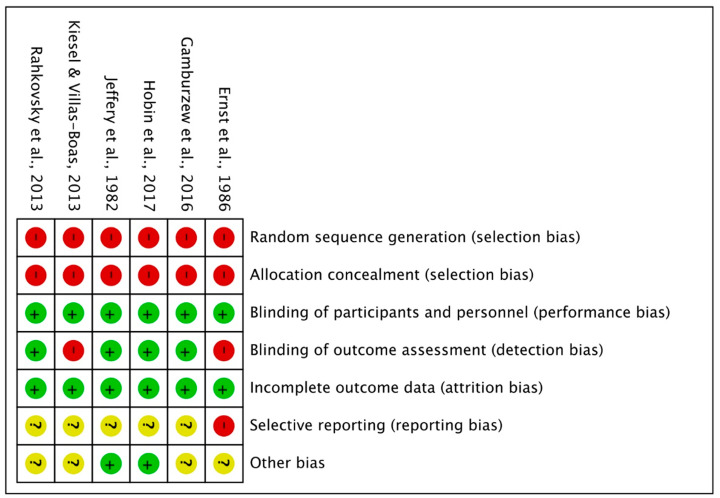
Risk of bias summary: review of the authors’ judgments about each. Risk of bias item for CBA studies (*n* = 6).

**Figure 4 foods-10-00922-f004:**
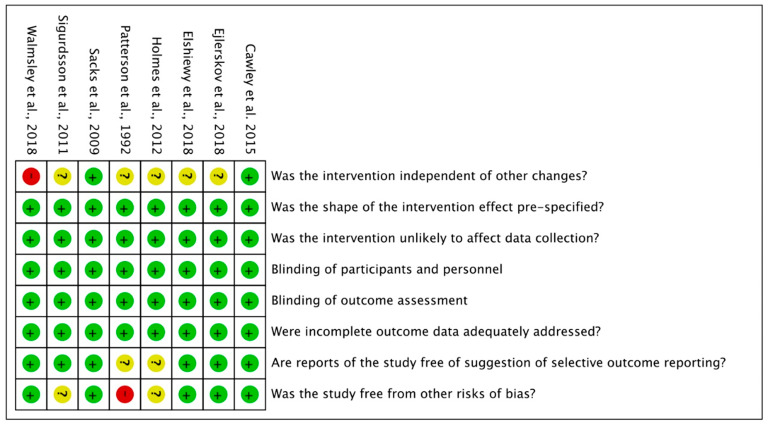
Risk of bias summary: review of the authors’ judgments about each. Risk of bias item for ITS studies (*n* = 8).

**Figure 5 foods-10-00922-f005:**
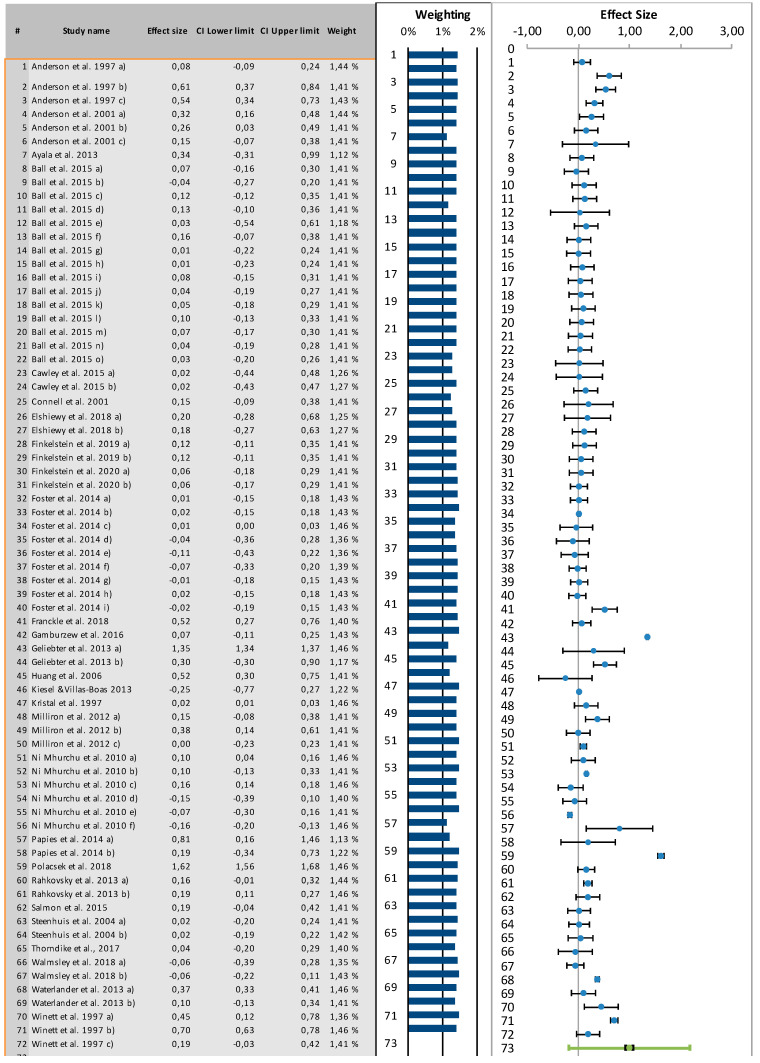
Results of meta−analysis: Forest plot of included studies results and effect size for in−store interventions on food purchase behavior (*k* = 72).

**Table 1 foods-10-00922-t001:** Eight strategies to promote healthy food and beverage environments in grocery stores. Adaption version of Kraak et al. [41] marketing and choice architecture framework.

Strategy	Description
Portioning	Reduce and/or standardize the portion size of food and beverage products that meet recommended nutrient targets to influence customers’ expectations about single servings and appropriate portions to support healthy dietary guidelines.
Place	Changing the internal setting (e.g., lighting, smell, music and branding of stores) that impact the ambience or atmospherics to highlight healthy food and beverage products.
Proximity	Placing healthier products at eye level or physically closer to customers at point-of-choice and point-of-purchase (e.g., placing healthier options at the entry or exit of store and giving healthy options better placement in the shelf).
Promotion	Use of marketing practices inside store that support healthier diets (i.e., products samples, taste-testing, in-store demonstrations, inside store audio public service announcements and education sessions inside store to promote healthy products).
Healthy Default Picks	Use of environmental cues that convenient, accepted and expected to socially normalize healthy defaults choices (e.g., introducing swaps that offer customers the opportunity to replace their usual food with healthier alternatives).
Pricing	Use of pricing strategies to increase sales of products that meet recommend nutrient targets to support healthy dietary guidelines (e.g., changes in price per unit, coupons and cash-back).
Prompting	Use of information on products to help customers make healthier choices at point-of choice and point-of-purchase (e.g., guiding star labeling system, nutrition labels and traffic-light labels).
Profile	Change in the product’s nutritional profile, quality, smell, taste, texture, flavor of food or beverage products that make meeting nutritional targets according to dietary guidelines.

**Table 2 foods-10-00922-t002:** Summary of the narrative synthesis of in-store interventions aimed at improving food and beverage purchases concerning author, source, year, country, setting, study design, intervention type, target product, study duration, store number, participant number, targeted population group and outcomes measured.

Reference	Source	Country	Setting	Study Design	Intervention Type	Target Product	Study Duration	Store Number	Participant Number	Targeted Population Type	Outcome Measurement
Achabal et al. [60]	Van ‘t Riet [24] + Hartman et al. [30] + Cadario and Chandon [15] + Cameron et al. [29] + Liberato et al. [31]	USA	Grocery store	RCT	Treatments assigned were: control group—no intervention prompting nr. 1–nutritional label showing “key nutrients” of the product, selection advice and calorie content and picture of food item prompting nr 2.—information label showing selection advice and picture of food item.	Increase volume sales of six targeted fruit and vegetable products (e.g., carrots, broccoli, cabbage, cauliflower, kiwi, tomatoes)	4 weeks baseline, 4 weeks intervention and 4 weeks follow-up	124 control stores and 248 intervention stores	283, not specified how many in each group	Normal population	Sales data
Anderson et al. [61] *	Database search + Adam and Jensen [28] + Afshin et al. [87] + Liberato et al. [31] + Hartman et al. [30]	USA	Grocery store	RCT	Treatment part of the American Nutrition for a Lifetime System program administered via kiosks in supermarkets. Treatments assigned were: control group—no intervention promotion and pricing—15 kiosk segments giving customers nutritional advice. All segments took about 10–20 min to complete. Pricing strategies include weekly coupons given to participants through kiosk ranked from 8 to 12 dollars.	Reduce grams of fat, increase grams of fiber per 1000 kcal purchased and increase servings of fruits and vegetables sold	10 weeks baseline and 14 weeks intervention	2 intervention stores	Treatm. *n* = 54 Control *n* = 50	Normal population	Self-reported
Anderson et al. [62] *	Database + Cameron et al. [29]	USA	Grocery store	RCT	Treatment part of the Nutrition for a Lifetime System program administered via kiosks in supermarkets. Treatments assigned were: control group—no intervention promotion and pricing—15 kiosk segments giving customers nutritional advice. All segments took about 10-20 min to complete. Pricing strategies include weekly coupons given to participants through kiosk ranked from 8 to 12 dollars.	Reduction percent of calories from total fat sold. Increase in percent calories from fiber and in servings of fruits and vegetables sold.	4 weeks baseline and 15 weeks intervention	5 intervention stores	Treatm. *n* = 145 Control *n* = 121	Normal population	Sales data
Ayala et al. [52] *	Database search + Adam and Jensen [28] + Liberato et al. [31]	USA	Grocery store	RCT	Treatments assigned were: control group—no intervention promotion and proximity—Promotion aim was to give nutritional education to customers included food demonstrations, distribution of recipe with healthy messages and an audio-based marketing campaign within store. Proximity included building a buffet bar with ready-to-eat fruits and vegetables, thereby increasing proximity of healthy options.	Increase servings of fruits and vegetables sold.	3 weeks baseline, 8 weeks intervention and 2 weeks follow-up	2 control stores and 2 intervention stores	Treatm. *n* = 61 Control *n* = 58	Low-income neighborhood	Self-reported
Ball et al. [63] *	Database search + Adam and Jensen [28] + Hartman et al. [30]	Australia	Grocery store	RCT	Treatment part of the Australian Supermarket Healthy Eating for Life program. Treatments assigned were: control group—no intervention pricing—price discounts of 20% on targeted products promotion—aiming at education customers about nutrition through 8 newsletters with recipes accompanying behavior-change and supplementary resources (including activities such as budgeting worksheets, goal-setting and self-monitoring exercises) pricing and promotion (as described above)	Increase grams of fruits and vegetables sold separately (g/week) and increase in milliliters of water and diet beverage sold (mL/week). Reduction in sugar-sweetened beverage (mL/week) sold.	12 weeks baseline, 12 weeks intervention and 24 weeks follow-up	2 intervention stores	Treatm. *n* = 574 Control *n* = 147	Normal population	Sales data
Cawley et al. [74] *	Database search + Cadario and Chandon [15] + Cameron et al. [29]	USA	Grocery store	ITS	Treatments assigned were: before condition—no intervention prompting—guiding stars labeling. Nutrition rating system on store shelves rating products with no-star, one-star, two-stars and three stars.	Increase in total volume sales of product categorized as healthy and reduction in products categorized as unhealthy according to the guiding star labeling system. Total of 102 products.	40 weeks baseline, 12 weeks intervention and 104 weeks follow-up	168 intervention stores	Treatm. *n* = 38,303 Control *n* = 335,120	Normal population	Sales data
Connell et al. [64] *	Glanz et al. [26] + Liberato et al. [31] + Escaron et al. [25] + Cameron et al. [29]	USA	Grocery store	RCT	Treatments assigned were: control group—no intervention promotion—in-store public service announcement every 30 min about aimed at increasing nutritional knowledge of customers. Intervention also included an outside-store component, which were 2 take-home audiotapes that customers were asked to play within the 4 intervention weeks. Additionally, included promotion outside store: Take-home audiotape (2 each with 1 h program) about nutritional knowledge about the importations of fruits and vegetable consumption.	Increase percentage average intake of fruits and vegetables	4 weeks intervention	3 control stores and 3 intervention stores	Treatm. *n* = 354 Control *n* = 328	Normal population	Self-reported
Ejlerskov et al. [88]	Database search	UK	Grocery store	ITS	Treatments assigned were: control group—no change in checkout policy proximity—“healthy checkout” meaning that unhealthy products such as sweets and chocolate at checkout were replaced with healthier options such as dried fruit, nuts, juices and water.	Reduction in volume of common unhealthy checkout-foods (sugary confectionary, chocolate and potato crisps) sold	52 weeks baseline and 52 weeks intervention	3 control stores and 6 intervention stores	Treatm. *n* = 30,000	Normal population	Sales data
Elshiewy et al. [85] *	Database search	UK	Grocery store	ITS	Treatments assigned were: before condition—no intervention prompting—nutritional label that displays the number of calories and the number of sugars, fat, saturated-fat and salt in grams per serving	Reduction in volume of unhealthy snacks (cookies), high sugar breakfast cereals and unhealthy beverage (soft drinks)	52 weeks baseline and 52 weeks intervention	2000 intervention stores	Treatm. *n* = 188,062	Normal population	Sales data
Ernst et al. [75]	Cameron et al. [29] + Escaron et al. [25] + van ’t Riet [24] + Liberato et al. [31]	USA	Grocery store	CBA	Treatment part of the American Nutrition for a Lifetime System program called “Eat for Health” administered by the American National Cancer Institute. Treatments assigned were: before condition—no intervention promotion—marketing healthy products through brochures, posters, labels next to food products with information and shelf-labels. Additionally, included promotion outside store: newspaper, advertisements and radio announcements focusing on healthy eating.	Increase in total volume sales of product categorized as healthy and reduction in products categorized as unhealthy. Total of 246 products.	52 weeks intervention	10 control stores and 10 intervention stores	Treatm. *n* = 1202 Control *n* = 1197	Normal population	Self-reported
Finkelstein et al. [71] *	Database search	Singapore	Online grocery store	RCT	Treatments assigned were: control group—no intervention prompting nr. 1—“Nutri-Score” label that gives information on the “grade” of products obtained according to the overall nutritional quality (rating from A to E). The different grades have different colors and A is green, C is yellow, and E is red prompting nr. 2—”Multiple Traffic Light” label gives information about calories, sugar, fat, saturated-fat, sodium content of each product and colors the different categories according to nutritional quality for each specific category.	Increase in total volume sales of product categorized as healthy according to the healthy eating Index (AHEI-2010)	3 weeks intervention	1 intervention stores	Total sample = 147	Normal population	Sales data
Finkelstein et al. [86] *	Database search	Singapore	Online grocery store	RCT	Treatments assigned were: control group—no intervention prompting nr. 1— “Lowest calories” label—labeling the 20% lowest calories per serving products within each product category prompting nr. 2—“Lowest calories” label—labeling the 20% lowest calories per serving products across all product categories	Reduction in total sales of calories purchased across all products sold in store.	3 weeks intervention	1 intervention stores	Total sample = 146	Normal population	Sales data
Foster et al. [56] *	Database search + Cadario and Chandon [15] + Adam and Jensen [28] + Liberato et al. [31] + Hartman et al. [30] + Cameron et al. [29]	USA	Grocery store	RCT	Treatments assigned were: control group—no intervention promotion and proximity—promotion included signs with the recommended product’s, shelf runners, taste-testing and free samples of recommended products. Some products were also bundled together (only for cereal and healthy beverages). Proximity included the increased number of facings and better placement of the recommended products	Increase in overall sales of products categorizes as healthy (skim milk, water, cheerios cereal and honeycomb cereal, diet Pepsi) and reduction in sales of products categorizes as less healthy (1% fat milk, Pepsi, Aquafina and water)	12 weeks baseline and 24 weeks intervention	4 control stores and 4 intervention stores	Treatm. *N* = 562,247 Control *n* = 635,028	Low-income neighborhood	Sales data
Franckle et al. [55] *	Database search	USA	Grocery store	RCT	Treatments assigned were: control group—monthly nutrition letters with general nutritional information prompting and pricing—all beverage products were labeled as red, yellow or green according to sugar-content. Participants received a gift card of 25 dollars per month for not purchasing the red-labeled beverage. A gift-card was sent with a monthly letter to the intervention group.	Reduction in volume sales of units of unhealthy beverage (sugar soda)	8 weeks baseline and 16 weeks intervention	1 intervention stores	Treatm. *N* = 71 Control *n* = 77	Low-income neighborhood	Sales data
Gamburzew et al. [57] *	Database search + Cadario and Chandon [15]	France	Grocery store	CBA	Treatments assigned were: before condition—no intervention promotion and proximity—promotion included posters, shelf-labels and taste-testing boots. Proximity strategy included placing targeted products at arm/eye level and in the middle of the shelf.	Increase in total volume sales of product categorized as healthy.	24 weeks baseline and 24 weeks intervention	2 control stores and 2 intervention stores	Treatm. *N* = 2651 Control *n* = 3974	Low-income neighborhood	Sales data
Geliebter et al. [59] *	Database search + Adam and Jensen [28] + Hartman et al. [30]	USA	Grocery store	RCT	Treatments assigned were: control group—no intervention pricing—discount of 50% for fruits, vegetables, bottled water and diet soda	Increase in total volume sales of fruits and vegetables	4 weeks baseline, 8 weeks intervention and 4 weeks follow-up	2 intervention stores	Treatm. *N* = 19 Control *n* = 28	Targeting overweight or obesity customers	Sales data
Hobin et al. [72]	Database search	Canada	Grocery store	CBA	Treatments assigned were: before group—no intervention prompting—Guiding Stars labeling. Nutrition rating system on store shelves rating products with no-star, one-star, two-stars of three stars.	Increase in total volume sales of product categorized as healthy (e.g., products with 3-stars, according to the Guiding Star labeling system)	12 weeks baseline and 36 weeks intervention	82 control stores and 44 intervention stores	n.a.	Normal population	Sales data
Holmes et al. [76]	Database search + Adam and Jensen [28] + Escaron et al. [25] + Cameron et al. [29]	USA	Grocery store	ITS	Treatments assigned were: before group—no intervention promotion—healthy campaign through the kiosk inside store showing 32 food products and poster of the food pyramid, including taste-testing booths and recipes.	Increase in proportion of total volume sales from 32 products categorized as healthy	5 weeks baseline, 12 weeks intervention and 5 weeks follow-up	1 intervention stores	Treatm. *N* = 112,072 Control (before) *n* = 46,960	Normal population	Sales data
Huang et al. [83] *	Liberato et al. [31] + Hartman et al. [30] + Cameron et al. [29]	Australia	Online grocery store	RCT	Treatments assigned were: control group—received non-specific advice about how to choose a diet lower in saturated-fat Healthy defaults through giving customers option to either retain the chosen product in the basket or swap products with an alternative lower in saturated-fat	Reduction in grams of saturated-fat (g/100g) sold	5 weeks intervention	1 intervention stores	Treatm. *N* = 251 Control *n* = 246	Normal population	Sales data
Jeffery et al. [80]	Database search + Escaron et al. [25] + van ’t Riet [24] + Liberato et al. [31] + Hartman et al. [30] + Cameron et al. [29]	USA	Grocery store	CBA	Treatments assigned were: before condition—no intervention promotion—marketing healthy diets educational posters about benefits of low fat diets, recipe cards and brochures in the dairy section.	Increase in volume sales of 25 low fat dairy products	16 weeks baseline and 24 weeks intervention	4 control stores and 4 intervention stores	n.a.	Normal population	Sales data
Kiesel & Villas-Boas [81]	Database search + Cadario and Chandon [15]	USA	Grocery store	CBA	Treatments assigned were: control group—no intervention prompting—nutritional shelf-label on 93 healthy popcorn options. 4 different combinations of labels with claims; “low fat”; “low fat” and “low calorie”; “low fat“, “low calorie” and “no trans-fat” and “low fat” with text about FDA approval.	Increase in volume sales of 93 low-calorie microwave popcorn products	3 weeks baseline and 3 weeks intervention	27 control stores and 5 intervention stores	Treatm. *N* = 742 Control *n* = 1080	Normal population	Sales data
Kristal et al. [65] *	Database search + Epstein et al. [89] + Escaron et al. [25] + Liberato et al. [31] + Hartman et al. [30] + Cameron et al. [29]	USA	Grocery store	RCT	Treatments assigned were: control group—no intervention promotion and price—flyers with information about fruits and vegetables on sale (reduced price), recipes and menus using the sale item and food demonstrations. Fifty cent coupons for all fruits and vegetables.	Increase in grams sold of fruits and vegetables by total weight (fresh, frozen and dried) or volume (canned)	0 weeks baseline and 52 weeks intervention	4 control stores and 4 intervention stores	Treatm. *N* = 356 Control *n* = 371	Normal population	Self-reported
Mhurchu et al. [84] *	Database search + Adam and Jensen [28] + [26] Glanz et al. [26] + Liberato et al. [31] + Epstein et al. [89] + Hartman et al. [30]	New Zealand	Grocery store	RCT	Treatments assigned were: control group—no intervention promotion—in-person nutrition education within the store. Educations recommended brand-specific healthier alternatives to less healthy foods usually purchased price 12.5% discount on healthy foods promotion and price (as described above)	Reduction in percentage calories sold from saturated-fat	24 weeks baseline, 24 weeks intervention and 24 week follow-up	12 control stores and 12 intervention stores	Treatm. *N* = 826 Control *n* = 278	Normal population	Sales data
Milliron et al. [66] *	Database search + Adam and Jensen [28] + Liberato et al. [31] + Hartman et al. [30] + Cameron et al. [29]	USA	Grocery store	RCT	Treatments assigned were: control group—no intervention promotion and promoting—in-person nutrition education within store, healthful shopping list and a monthly newsletter and recipes were also available in stores. Prompting were nutrition shelf-labels identifying products as healthy options.	Increase in volume sales of saturated-fat (g/1000 kcals), and servings of fruits, vegetables (servings/1000 kcals)	16 weeks baseline and 16 weeks intervention	1 intervention stores	Treatm. *N* = 70 Control *n* = 83	Normal population	Self-reported
Papies et al. [58] *	Cameron et al. [29]	Netherlands	Grocery store	RCT	Treatments assigned were: control group—flyer with new recipe card with no diet prime promotion—flyer with low-calorie recipe card with diet prime (healthy, good for your figure and number of calories)	Reduction in volume sales of units of unhealthy snacks sold (e.g., sum of units of cake cookies, sweets chocolate, chips and other savory and nut snacks purchased)	n.a.	1 intervention stores	Treatm. *N* = 49 Control *n* = 50	Targeting overweight or obesity customers and normal population	Sales data
Patterson et al. [77]	Escaron et al. [25] + van ’t Riet [24] + Cameron et al. [29]	USA	Grocery store	ITS	Treatment part of the American Nutrition for a Lifetime System program called “Eat for Health” administered by the American National Cancer Institute. Treatments assigned were: before condition—no intervention promotion—promote healthy products through brochures, posters, labels next to food products with information and shelf-labels. Additionally, included promotion outside store: newspaper, advertisements and radio announcements focusing on healthy eating.	Increase in total volume sales of product categorized as healthy and reduction in products categorized as unhealthy. Total of 8 food categories	52 weeks baseline 24 and 104 weeks intervention	20 control stores and 20 intervention stores	*n* = 1,920,000	Normal population	Sales data
Polacsek et al. [54] *	Database search	USA	Grocery store	RCT	Treatments assigned were: control group—no intervention pricing—50% discount on fruits and vegetables	Increase in overall dollars spent on fruits and vegetables	12 weeks baseline and 16 weeks intervention	1 intervention stores	Treatm. *N* = 183 Control *n* = 171	Low-income neighborhood	Sales data
Rahkovsky et al. [73] *	Database search	USA	Grocery store	CBA	Treatments assigned were: before condition—no intervention prompting—guiding stars labeling system. Support material in-store included brochures, signs and kiosks explaining the prompt.	Increase in total volume sales of product categorized as healthy (e.g., products with 3-stars). Reduction in total volume sales of product categorized as unhealthy (e.g., unstarred).	52 weeks baseline and 88 weeks intervention	134 intervention stores	After *n* = 11,658 Before *n* = 7102	Normal population	Sales data
Sacks et al. [78]	Hersey et al. [27]	United Kingdom	Grocery store	ITS	Treatments assigned were: before condition—no intervention promoting—front-of-package of traffic-light labeling	Increase in volume sales of product categorized as healthy (e.g., green-labeled). Reduction in volume sales of product categorized as unhealthy (red-labeled). Product categories were chilled prepackaged meals (ready meals) and fresh prepackaged sandwiches products.	4 weeks baseline and 4 weeks intervention	1 intervention stores	After *n* = 11,658 Before *n* = 7102	Normal population	Sales data
Salmon et al. [79] *	Adam and Jensen [28] + Cameron et al. [29]	Netherlands	Grocery store	RCT	Treatments assigned were: control group—no intervention promotion—marketing “low fat” by using shelf banner with slogan “most sold in this supermarket”	Increase in volume sales in one healthier low fat cheese	4 days baseline and 4 days intervention	1 intervention stores	Treatm. *N* = 32 Control *n* = 41	Normal population	Sales data
Sigurdsson et al. [67]	Cameron et al. [29]	Norway	Grocery store	ITS	Treatments assigned were: before period—no intervention proximity—placing bananas at checkout proximity—placing bananas at sweet shelf proximity and promotion—placing bananas at checkout and sweet shelf and advertising bananas in store.	Increase sales volume of banana as proportion of total sales of fruits.	4 weeks baseline and 4 weeks intervention	1 intervention stores	n.a.	Normal population	Sales data
Steenhuis et al. [82] *	Database search	Netherlands	Grocery store	RCT	Treatments assigned were: control group—no intervention promotion—marketing reduced fat intake program through posters with information about the program, a brochure about healthy eating, recipe cards, and a self-help manual. promotion and prompting—promotion as described above combined with a nutritional shelf-labeling system. Healthier options in nine food categories were labeled with program logo, name of item, indication that it was a good low fat choice.	Reduction in participants average fat intake	4 weeks baseline and 24 weeks intervention	4 control stores and 9 intervention stores	Treatm. *N* = 1464 Control *n* = 739	Normal population	Self-reported
Thorndike et al. [53] *	Hartman et al. [30]	USA	Grocery store	RCT	Treatments assigned were: control group—no intervention proximity and profile—placing fruits and vegetable in new shelf racks, placing fruits and vegetables in front of stores in new baskets. Profile of products was improved by consulting store owners about which products to stock and how to identify items that are starting to go bad.	Increase in volume sales of fruits and vegetables	44 weeks baseline 22, weeks intervention and 20 week follow-up	3 control stores and 3 intervention stores	Treatm. *N* = 295 Control *n* = 280	Low-income neighborhood	Self-reported
Walmsley et al. [68] *	Database search	United Kingdom	Grocery store	ITS	Treatments assigned were: control group—no intervention proximity—moving fruit and vegetables from the back of the store to the aisle closest to the entrance. Entrance-facing display of fruits and vegetables were replaced with chiller cabinets.	Increase in volume sales of fruits and vegetables	90 weeks baseline and 80 weeks intervention	1 intervention stores	Treatm. *N* = 5464 Control *n* = 5790	Normal population	Sales data
Waterlander et al. [69] *	Database search + Afshin et al. [87] + Adam and Jensen [28] + Liberato et al. [31] + Hartman et al. [30]	Netherlands	Grocery store	RCT	Treatments assigned were: control group—no intervention price—price discount of 50% for fruits and vegetables through coupons. Price and promotion outside store. Price discount of 50% for fruits and vegetables through coupons. Promotion was by giving customers education about energy content of foods through recipes and books.	Increase in volume sales of fruits and vegetables (kg per household for 2 week intervals)	2 weeks baseline, 24 weeks intervention and 24 weeks follow-up	1 control stores and 3 intervention stores	Treatm. *n* = 115 Control *n* = 36	Normal population	Sales data
Winett et al. [70] *	van ’t Riet [24] + Hartman et al. [30] + Cameron et al. [29] + Cadario and Chandon [15] + Liberato et al. [31]	USA	Grocery store	RCT	Treatment part of a modified version of the Nutrition for a Lifetime System program administered via kiosks in supermarkets. Treatments assigned were: control group—no intervention promotion and pricing—10 kiosk segments giving customers nutritional advice. All segments took about 2-10 min to complete. Pricing strategies include weekly coupons given to participants through the kiosk ranked from 0.5 to 1 dollar.	Reduction in % intake of calories from total fat, increase in gram fiber and increase in servings of fruits and vegetables	10 weeks intervention and 4 weeks follow-up	2 intervention stores	Treatm. *n* = 54 Control *n* = 51	Normal population	Sales data

Comment: * article included in meta-analysis.

**Table 3 foods-10-00922-t003:** Effect size (ES) and confidence interval (95%) for different in-store interventions, *k* = 72.

Intervention	Frequency (k)	ES (d)	CI Lower Limit	CI Higher Limit
Promotion and pricing	17	0.21 *	0.08	0.33
Promotion	15	0.10 *	0.02	0.18
Promotion and proximity	12	0.01 *	0.00	0.02
Pricing	11	0.40 *	0.00	0.80
Prompting	12	0.14 *	0.09	0.19
Pricing and prompting	1	0.52 *	0.27	0.76
Promotion and prompting	1	0.02	−0.20	0.24
Healthy default picks	1	0.52 *	0.30	0.75
Profile and Proximity	1	0.04	−0.20	0.29
Proximity	1	−0.06	−0.39	0.28

Comment. * significant ES.

**Table 4 foods-10-00922-t004:** Effect size (ES) and confidence interval (95%) for different target product categories, *k* = 72.

Target Product Category	Frequency (k)	ES (d)	CI Lower Limit	CI Higher Limit
Increase healthy products	46	0.19 *	0.09	0.29
Fruit and vegetables	21	0.28 *	0.08	0.48
Healthy beverage	10	0.01 *	0.01	0.02
Total volume healthy	8	0.16 *	0.14	0.17
Fiber	5	0.29	−0.17	0.76
Low-calorie snacks	1	−0.25	−0.77	0.27
Low-fat cheese	1	0.19 *	−0.04	0.42
Reduction less healthy products	26	0.11 *	0.03	0.19
Fat	7	0.19	−0.01	0.39
Unhealthy beverage	7	0.09	−0.07	0.25
Total volume unhealthy	6	−0.03	−0.19	0.12
Calorie	3	0.07	−0.06	0.20

Comment. * significant ES.

**Table 5 foods-10-00922-t005:** Quality of evidence and summary of findings.

Certainty Assessment							Summary of Findings
N^o^ of Studies	Risk of bias	Inconsistency	Indirectness	Imprecision	Publication Bias	Overall Certainty of Evidence	SMD Effect Size 95% CI
RCT studies							
22	Very serious	Serious	Not serious	Serious	None	⨁◯◯◯ Very low	0.18 [0.10, 0.26]
CBA studies							
6 *	Serious	Not serious	Not serious	Serious	NA ***	Low	0.14 [−0.01, 0.30]
ITS studies							
8 **	Not serious	Not serious	Not serious	Not serious	NA ***	Low	−0.01 [−0.11, 0.09]

Comment: * = 3 studies included in ES, ** = 3 studies included in ES, *** = Fewer than 10 studies included.

## Data Availability

The data supporting the conclusions in this article are included within the submitted Appendix A.

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
