# Peer review of "Efficiency of In-Store Interventions to Impact Customers to Purchase Healthier Food and Beverage Products in Real-Life Grocery Stores: A Systematic Review and Meta-Analysis"

_foods, 2021, doi:10.3390/foods10050922_

Round 1

Reviewer 1 Report

The paper aims to review the effectiveness of interventions in grocery stores and how they impact sales. This is a well-written paper, that follows well-regarded methodology. Overall, I think the paper could be improved by including more discussion about how intervention can lead to healthier purchasing habits. Other comments are listed below. I apologize for only listing the page number in some cases, but there were no line numbers.

Page 9- “The Guiding State Rating System…”, was evaluated in three studies. Which studies? This should be identified.

Page 9- “Eight studies targeted reduction in unhealthy food products”, how did these studies define unhealthy food products? And was the definition similar for all eight studies? Also, in the statement after, how did the studies define healthy products?

Page 21, Line 1- In the word figure, the “f” should be capitalized.

Page 21, Line 26- What do the authors mean by “One study high risk of other biases because analysis deviation from the study protocol”?

Page 27, Line 194- Why do the authors think that not many studies looked at combinations of interventions?

Page 30, Line 297- “Further research is also needed” is said repeatedly in this section.

Reviewer 2 Report

This is a very relevant and good written paper of a well performed review research. The work is very well done. I only have a few minor issues. The most important one is the way results are discussed. The paper shows very clearly that the effect of all the different methods to persuade consumers in store do not seem to work, or only in a very limited way. It would be good if this conclusion should be prominent in the paper. We see this sort of results all the time and still trust is based on this kind of methods e.g .by governments to persuade consumers, in stead of coming up with strict rules like high taxes on unhealthy food, a ban on selling tabac in supermarkets, et. 

It all simple means that without strong financial or governmental strategies it will be very hard to persuade consumers to buy "better" food. This is even stronger if we take into account the biases of the research as indicated by the authors themselves. 

The second issue is the relatively low amount of papers that are included in the review. Maybe some extra attention can be given to the reason why only this limited amount of papers survives the quality control. 

Some papers are added to the analysis, however, without reporting all the info that is needed. Why are these papers included, and what happens to the results if these papers are excluded?

Some sentences are a bit unclear, probably due to editing. Please do a last close reading on the manuscript.  
